# How Faithful is your Synthetic Data? Sample-level Metrics for Evaluating and Auditing Generative Models

## Abstract

Devising *domain-* and *model-agnostic* evaluation metrics for generative models is an important and as yet unresolved problem. Most existing metrics, which were tailored solely to the image synthesis setup, exhibit a limited capacity for diagnosing the modes of failure of generative models across broader application domains. In this paper, we introduce a 3-dimensional metric, *($\alpha$-Precision, $\beta$-Recall, Authenticity)*, that characterizes the *fidelity*, *diversity* and *generalization* performance of *any* generative model in a domain-agnostic fashion. Our metric unifies statistical divergence measures with precision-recall analysis, enabling sample-level and distribution-level diagnoses of model fidelity and diversity. We introduce *generalization* as an additional dimension for model performance that quantifies the extent to which a model copies training data—a crucial performance indicator when modeling sensitive data with requirements on privacy. The three metric components correspond to (interpretable) probabilistic quantities, and are estimated via sample-level binary classification. The sample-level nature of our metric inspires a novel use case which we call *model auditing*, wherein we judge the quality of individual samples generated by a (black-box) model, discarding low-quality samples and hence improving the overall model performance in a post-hoc manner.

## 1 Introduction

Intuitively, it would seem that evaluating the likelihood function of a generative model is all it takes to assess its performance. As it turns out, the problem of evaluating generative models is far more complicated. This is not only because state-of-the-art models, such as Variational Autoencoders (VAE) (Kingma & Welling (2013)) and Generative Adversarial Networks (GANs) (Goodfellow et al. (2014)), do not possess tractable likelihood functions, but also because the likelihood score itself is a flawed measure of performance—it scales badly in high dimensions, and it obscures distinct modes of model failure into a single uninterpretable score (Theis et al. (2015)). Absent domain-agnostic metrics, earlier work focused on crafting domain-specific scores, e.g., Inception score (Salimans et al. (2016)), with an exclusive emphasis on image data (Lucic et al. (2018)).

In this paper, we introduce an alternative approach to evaluating generative models, where instead of assessing the generative distribution by looking at *all* synthetic samples *collectively* to compute likelihood or divergence, we classify *each* sample *individually* as being of high or low quality. In this way, our metric comprises interpretable probabilistic quantities—resembling those used to evaluate discriminative models (e.g., AUC-ROC)—which describe the rate by which a model makes errors. When averaged over all samples, our sample-level scores reflect discrepancy between real and generative distributions in a way similar to statistical divergence measures (e.g., KL divergence, Fréchet distance (Heusel et al. (2017)), or maximum mean discrepancy (Sutherland et al. (2016)). In this sense, our metric enables diagnosing model performance on both the sample and distribution levels.

Our metric represents the performance of a generative model as a point in a 3-dimensional space—each dimension corresponds to a distinct quality of the model. These qualities are: *Fidelity*, *Diversity* and *Generalization*. Fidelity measures the quality of a model's synthetic samples, and Diversity is

Figure 1: **Pictorial depiction for the proposed evaluation metrics.** The blue and red spheres correspond to the $\alpha$- and $\beta$-supports of real and generative distributions, respectively. Blue and red points correspond to real and synthetic data. (a) Synthetic data falling outside the blue sphere will look unrealistic or noisy. (b) Overfitted models can generate ostensibly high-quality data samples that are "unauthentic" because they are copied from the training data. (c) High-quality data samples should reside inside the blue sphere. (d) Outliers do not count in the $\beta$-Recall metric. (Here, $\alpha=\beta=0.9$, $\alpha$-Precision=8/9, $\beta$-Recall = 4/9, and Authenticity = 9/10.)

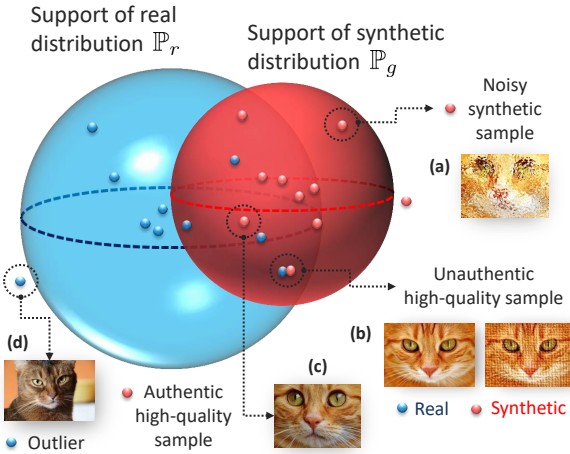

the extent to which these samples cover the full variability of real samples, whereas Generalization quantifies the extent to which a model *overfits* (copies) training data.

**How do we quantify the 3 dimensions of performance?** We build on the precision-recall analysis framework proposed in (Sajjadi et al. (2018)), and introduce the $\alpha$-Precision and $\beta$-Recall metrics to quantify model Fidelity and Diversity, respectively. Both metrics assume that a fraction $1 - \alpha$ (or $1 - \beta$) of the real (and synthetic) data are "outliers", and $\alpha$ (or $\beta$) are "typical". $\alpha$-Precision is the fraction of synthetic samples that resemble the "most typical" $\alpha$ real samples, whereas $\beta$-Recall is the fraction of real samples covered by the most typical $\beta$ synthetic samples. $\alpha$-Precision and $\beta$-Recall are evaluated for all $\alpha, \beta \in [0, 1]$, providing entire precision and recall curves instead of single numbers. To compute both metrics, we embed the (real and synthetic) data into hyperspheres with most samples concentrated around the centers, i.e., the real and generative distributions ($\mathbb{P}_r$ and $\mathbb{P}_g$) has spherical-shaped supports. In this transformed feature space, typical samples would be located near the centers of the spheres, whereas outliers would be closer to the boundaries.

To quantify Generalization, we introduce the *Authenticity* metric, which reflects the likelihood of a synthetic sample being copied from training data. We derive the Authenticity metric from a hypothesis test for data copying based on the observed proximity of synthetic samples to real ones in the embedded feature space. A pictorial illustration for all metrics is shown in Figure 1.

**How is our metric different?** If one think of standard precision and recall metrics as "hard" binary classifiers of real and synthetic samples, our $\alpha$-Precision and $\beta$-Recall can be thought of as soft-boundary classifiers that do not only compare the supports of $\mathbb{P}_r$ and $\mathbb{P}_g$, but also assesses whether both distributions are calibrated. Precision and recall metrics are special cases of $\alpha$-Precision and $\beta$-Recall for $\alpha = \beta = 1$. As we show later, our new metric definitions solve many of the drawbacks of standard precision-recall analysis, such as lack of robustness to outliers and failure to detect distributional mismatches (Naeem et al. (2020)). They also enable detailed diagnostics of different types of model failure, such as mode collapse and mode invention. Moreover, optimal values of our metrics are achieved only when $\mathbb{P}_r$ and $\mathbb{P}_g$ are identical, thereby eliminating the need to augment the evaluation procedure with additional measures of statistical divergence (e.g., KL divergence).

Previous works relied on pre-trained embeddings (using ImageNet feature extractors (Deng et al. (2009))). In this work, we propose feature embeddings that are model- and domain-agnostic, and are tailored to our metric definitions and data set at hand. Our proposed feature embedding step can be completely bespoke to raw data, or augmented with pre-trained embeddings. This enables our metric to remain operable in application domains where no pre-trained representations exist.

Overfitting is a crucial mode of failure of generative models, especially when modeling sensitive data (e.g., clinical data) for which data copying may violate privacy requirements (Yoon et al. (2020)), but it has been overlooked in previous works which focused exclusively on quantifying the Fidelity-Diversity characterization (Brock et al. (2018)). As we show in our experiments (Section 5), because our metric accounts for Generalization, it can provide a fuller picture of a generative model's performance. Precisely, we show that some of the celebrated generative models score highly for Fidelity

and Diversity simply because they memorize real samples, rendering them inappropriate for privacy-sensitive applications. A comprehensive survey of prior work is provided in the Appendix.

**Model *auditing* as a novel use case.** The sample-level nature of our metrics inspires the new use case of *model auditing*, wherein we judge individual synthetic samples by their quality, and reject samples that have low Fidelity or are unauthentic. In Section 5, we show that model audits can *improve* the outputs of a black-box model in a post-hoc fashion without any modifications to the model itself, and demonstrate this use case in synthesizing clinical data for COVID-19 patients.

## 2 EVALUATING AND AUDITING GENERATIVE MODELS

### 2.1 PROBLEM SETUP

We denote real and generated data as $X_r \sim \mathbb{P}_r$ and $X_g \sim \mathbb{P}_g$, respectively, where $X_r, X_g \in \mathcal{X}$, with $\mathbb{P}_r$ and $\mathbb{P}_g$ being the *real* and *generative* distributions, and $\mathcal{X}$ being the *input space*. The generative distribution, $\mathbb{P}_g$, is estimated using a generative model (e.g., GAN). The real and *synthetic* data sets are $\mathcal{D}_{real} = \{X_{r,i}\}_{i=1}^n$ and $\mathcal{D}_{synth} = \{X_{g,j}\}_{j=1}^m$, where $X_{r,i} \sim \mathbb{P}_r$ and $X_{g,j} \sim \mathbb{P}_g$.

### 2.2 WHAT MAKES A GOOD SYNTHETIC DATA SET?

Our goal is to construct a metric $\mathcal{E}(\mathcal{D}_{real}, \mathcal{D}_{synth})$ for the quality of $\mathcal{D}_{synth}$ in order to **(i) evaluate** the performance of the underlying generative model $\mathbb{P}_g$, and **(ii) audit** the model outputs by discarding (individual) "low-quality" samples, thereby improving the overall quality of $\mathcal{D}_{synth}$. In order for $\mathcal{E}$ to fulfill the evaluation and auditing tasks, it must satisfy the following desiderata: **(1)** it should be able to **disentangle the different modes of failure** of $\mathbb{P}_g$ through **interpretable** measures of performance, and **(2)** it should be **sample-wise computable**, i.e., we should be able to tell if a given (individual) synthetic sample $X_g \sim \mathbb{P}_g$ is of a low quality.

Having outlined the desiderata for our sought-after evaluation metric, we now propose three qualities of synthetic data that the metric $\mathcal{E}$ should be able to quantify. Failure to fulfill any of these three qualities correspond to independent modes of failure of the model $\mathbb{P}_g$. These qualities are:

1. *Fidelity*—the generated samples resemble real samples from $\mathbb{P}_r$. A high-fidelity synthetic data set should contain "realistic" samples, e.g. visually-realistic images.

2. *Diversity*—the generated samples are diverse enough to cover the variability of real data, i.e., a model should be able to generate a wide variety of good samples.

3. *Generalization*—the generated samples should not be mere copies of the (real) samples in training data, i.e., models that overfit to $\mathcal{D}_{real}$ are not truly "generative".

In Section 3, we propose a *three-dimensional* evaluation metric $\mathcal{E}$ that captures all of the qualities above. Our proposed metric can be succinctly described as follows:

$$\mathcal{E} \triangleq (\underbrace{\alpha\text{-Precision}}_{Fidelity}, \underbrace{\beta\text{-Recall}}_{Diversity}, \underbrace{\text{Authenticity}}_{Generalization}). \tag{1}$$

The $\alpha$-Precision and $\beta$-Recall metrics are generalizations of the conventional notions of precision and recall used in binary classification analysis (Flach & Kull (2015)). Precision measures the rate by which the model synthesizes "realistic-looking" samples, whereas the recall measures the fraction of real samples that are covered by $\mathbb{P}_g$. The *authenticity* score measures the fraction of synthetic samples that are invented by the model and not copied from the training data.

### 2.3 EVALUATION AND AUDITING PIPELINES

Having provided a bird's-eye view of our proposed metric $\mathcal{E}$, we now briefly summarize the steps involved in the evaluation and auditing tasks. Since statistical comparisons of complex data types in the raw input space $\mathcal{X}$ are difficult, the evaluation pipeline starts by embedding $X_r$ and $X_g$ into a "meaningful" feature space through a representation $\Phi$, dubbed the *evaluation embedding*, and then computing $\mathcal{E}$ on the embedded features (see Figure 2(a)). In Section 4, we propose a representation learning approach to construct embeddings tailored to our metric and the data set at hand.

In the (post-hoc) model auditing task, we compute the sample-level metrics for each $X_{g,j}$ in $\mathcal{D}_{synth}$, and discard samples with low authenticity and/or precision scores, which results in a "curated" synthetic data set with an improved overall performance. When granted direct access to the model $\mathbb{P}_g$, the auditor serves as a rejection sampler that repeatedly draws samples from $\mathbb{P}_g$, only accepting ones with high precision and authenticity (Figure 2(b)). Model auditing is possible through our metrics as they can be used to evaluate the quality of individual synthetic samples; the same task cannot be carried out with statistical divergence measures that compare the overall real and generative distributions.

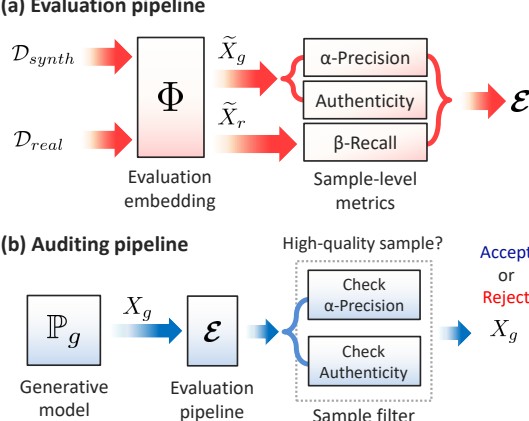

Figure 2: **The evaluation and auditing pipelines.**

# 3 $\alpha$-PRECISION, $\beta$-RECALL AND AUTHENTICITY

## 3.1 DEFINITIONS AND NOTATIONS

Let $\widetilde{X}_r = \Phi(X_r)$ and $\widetilde{X}_g = \Phi(X_g)$ be the *embedded* real and synthetic features. For simplicity, we will use $\mathbb{P}_r$ and $\mathbb{P}_g$ to refer to the distributions over the raw and embedded features interchangeably. Let $\mathcal{S}_r = \mathrm{supp}(\mathbb{P}_r)$ and $\mathcal{S}_g = \mathrm{supp}(\mathbb{P}_g)$, where $\mathrm{supp}(\mathbb{P})$ is the support of $\mathbb{P}$. Central to our proposed metrics is a more general notion for the support of a distribution $\mathbb{P}$, which we dub the $\alpha$-*support*. We define the $\alpha$-support as the *smallest* subset of $\mathcal{S} = \mathrm{supp}(\mathbb{P})$ supporting a probability mass $\alpha$, i.e.,

$$\mathcal{S}^\alpha \triangleq \min_{s \subseteq \mathcal{S}} V(s), \ s.t. \ \mathbb{P}(s) = \alpha, \tag{2}$$

where $V(s)$ is the volume (Lebesgue measure) of $s$, and $\alpha \in [0, 1]$. One can think of an $\alpha$-support as dividing the full support of $\mathbb{P}$ into "normal" samples concentrated in $\mathcal{S}^\alpha$, and "outliers" residing in $\bar{\mathcal{S}}^\alpha$, where $\mathcal{S} = \mathcal{S}^\alpha \cup \bar{\mathcal{S}}^\alpha$. The notion of $\alpha$-support is also known as the minimum volume set, and has been traditionally used in outlier detection models (Polonik (1997); Scott & Nowak (2006)).

Finally, we define the distance between a data sample $X$ and the training data $\mathcal{D}_{real}$ as the distance between $X$ and the closest sample in $\mathcal{D}_{real}$, i.e.,

$$d(X, \mathcal{D}_{real}) = \min_{1 \le i \le n} d(X, X_{r,i}), \tag{3}$$

where $d$ is a distance metric defined over the input space $\mathcal{X}$.

## 3.2 SAMPLE-LEVEL EVALUATION METRICS

### 3.2.1 $\alpha$-PRECISION AND $\beta$-RECALL

$\boldsymbol{\alpha}$**-Precision.** The conventional Precision metric is defined as the probability that a generated sample is supported by the real distribution, i.e. $\mathbb{P}(\widetilde{X}_g \in \mathcal{S}_r)$ (Sajjadi et al. (2018)). We propose a more refined measure of sample fidelity, dubbed the $\alpha$-*Precision* (denoted as $P_\alpha$), defined as follows:

$$P_\alpha \triangleq \mathbb{P}(\widetilde{X}_g \in \mathcal{S}_r^\alpha), \text{ for } \alpha \in [0, 1]. \tag{4}$$

That is, $P_\alpha$ is the probability that a synthetic sample resides in the $\alpha$-support of the real distribution.

$\boldsymbol{\beta}$**-Recall.** To assess diversity in synthetic data, we propose the $\beta$-Recall metric as a generalization of the conventional Recall metric. Formally, we define the $\beta$-Recall as follows:

$$R_\beta \triangleq \mathbb{P}(\widetilde{X}_r \in \mathcal{S}_g^\beta), \text{ for } \beta \in [0, 1], \tag{5}$$

i.e., $R_\beta$ is the fraction of real samples that reside in the $\beta$-support of the generative distribution.

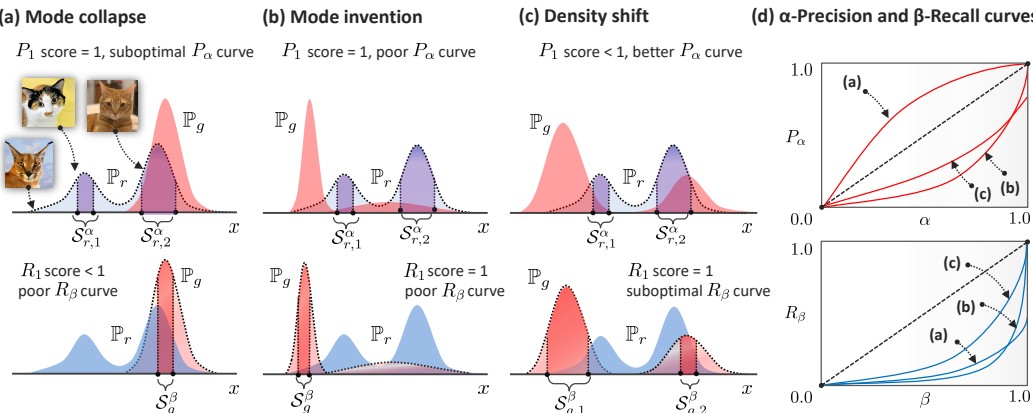

Figure 3: **Interpretation of the $P_\alpha$ and $R_\beta$ curves.** Real distribution is colored in blue, generative distribution is in red. Distributions are collapsed into 1 dimension for simplicity. Here, $\mathbb{P}_r$ is a multimodal distribution of cat images, with one mode representing orange tabby cats and another mode for Calico cats; outliers comprise exotic Caracal cats. Shaded areas represent the probability mass covered by $\alpha$- and $\beta$-supports—these supports concentrate around the modes, but need not be contiguous for multimodal distributions, i.e., we have $\mathcal{S}_r^\alpha = \mathcal{S}_{r,1}^\alpha \cup \mathcal{S}_{r,2}^\alpha$, and $\mathcal{S}_g^\beta = \mathcal{S}_{g,1}^\beta \cup \mathcal{S}_{g,2}^\beta$. **(a)** Here, the model $\mathbb{P}_g$ exhibits mode collapse where it over-represents orange tabbies. Such model would achieve a precision score of $P_1 = 1$ but a suboptimal (concave) $P_\alpha$ curve (panel (d)). Because it does not cover all modes, the model will have both a suboptimal $R_1$ score and $R_\beta$ curve. **(b)** This model perfectly nails the support of $\mathbb{P}_r$, hence it scores optimal standard metrics $P_1 = R_1 = 1$. However, the model invents a mode by over-representing outliers, where it mostly generates images for the exotic cat breed. Standard metrics imply that model (a) outperforms (b) where in reality (a) is more faithful to the real data. $P_\alpha$ and $R_\beta$ give us a fuller picture of the comparative performances of both models. **(c)** This model realizes both types of cats but estimates a slightly shifted support and density; intuitively, this is the best of the three models, but it will appear inferior to (b) under $P_1$ and $R_1$. By examining the $P_\alpha$-$R_\beta$ curves, we see that model (c) has less deviation from optimal performance (the dashed black lines in panel (d)).

**Interpreting $\alpha$-Precision and $\beta$-Recall.** To interpret (4) and (5), we first need to revisit the notion of $\alpha$-support. From (2), we know that an $\alpha$-support hosts the most densely packed probability mass $\alpha$ in a distribution, hence $\mathcal{S}_r^\alpha$ and $\mathcal{S}_g^\beta$ always concentrate around the modes of $\mathbb{P}_r$ and $\mathbb{P}_g$ (Figure 3); samples residing outside of $\mathcal{S}_r^\alpha$ and $\mathcal{S}_g^\beta$ can be thought of as outliers. In this sense, $P_\alpha$ and $R_\beta$ do not count outliers when assessing fidelity and diversity. That is, the $\alpha$-Precision score deems a synthetic sample to be of a high fidelity not only if it looks "realistic", but also if it looks "typical". Similarly, $\beta$-Recall counts a real sample as being covered by $\mathbb{P}_g$ only if it is not an outlier in $\mathbb{P}_g$. By sweeping the values of $\alpha$ and $\beta$ from 0 to 1, we obtain a varying definition of which samples are typical and which are outliers—this gives us entire $P_\alpha$ and $R_\beta$ curves as illustrated in Figure 3.

**Generalizing precision-recall analysis.** Unlike standard precision and recall, $P_\alpha$ and $R_\beta$ take into account not only the supports of $\mathbb{P}_r$ and $\mathbb{P}_g$, but also their densities. Standard precision (and recall) correspond to one point on the $P_\alpha$ (and $R_\beta$) curve; they coincide with $P_\alpha$ and $R_\beta$ evaluated on the full supports (i.e., $P_1$ and $R_1$). By defining our metrics with respect to the $\alpha$- and $\beta$-supports, we do not treat all samples equally, but assign higher importance to samples in "denser" regions of $\mathcal{S}_r$ and $\mathcal{S}_g$. $P_\alpha$ and $R_\beta$ reflect the extent to which $\mathbb{P}_r$ and $\mathbb{P}_g$ are *calibrated*— i.e., good $P_\alpha$ and $R_\beta$ are achieved when $\mathbb{P}_r$ and $\mathbb{P}_g$ share the same modes and not just a common support.

Our proposed $P_\alpha$ and $R_\beta$ metrics address major shortcomings of the commonly used $P_1$ and $R_1$, among these are: lack of robustness to outliers, failure to detect matching distributions, and inability to diagnose different types of distributional failure (Naeem et al. (2020)). Basically, $\mathbb{P}_g$ will score perfectly on precision and recall ($R_1=P_1=1$) as long as it nails the support of $\mathbb{P}_r$, even if $\mathbb{P}_r$ and $\mathbb{P}_g$ place totally different densities on their common support. Figure 3 illustrates how our metrics remedy these shortcomings. While optimal $R_1$ and $P_1$ are achieved by arbitrarily mismatched densities, our $P_\alpha$ and $R_\beta$ curves are optimized only when $\mathbb{P}_r$ and $\mathbb{P}_g$ are identical as stated by Theorem 1.

**Theorem 1.** *The $\alpha$-Precision and $\beta$-Recall satisfy the condition $P_\alpha/\alpha = R_\beta/\beta = 1$, $\forall \alpha, \beta$, if and only if the generative and real densities are identical, i.e., $\mathbb{P}_g = \mathbb{P}_r$.* ∎

That is, a model is optimal if and only if its $P_\alpha$ and $R_\beta$ are both straight lines with unity slopes.

**Measuring statistical discrepancy with $P_\alpha$ and $R_\beta$.** While the $P_\alpha$ and $R_\beta$ curves provide a detailed view on a model's fidelity and diversity, it is often more convenient to summarize performance in a single number. To this end, we define the mean absolute deviation of $P_\alpha$ and $R_\beta$ as:

$$\Delta P_\alpha = \int_0^1 |P_\alpha - \alpha|\, d\alpha, \ \ \Delta R_\beta = \int_0^1 |R_\beta - \beta|\, d\beta, \tag{6}$$

where $\Delta P_\alpha \in [0, 1/2]$ and $\Delta R_\beta \in [0, 1/2]$ quantify the extent to which $P_\alpha$ and $R_\beta$ deviate from their optimal values. We define the *integrated $P_\alpha$ and $R_\beta$* metrics as $IP_\alpha = 1 - 2\Delta P_\alpha$ and $IR_\beta = 1 - 2\Delta R_\beta$, both take values in $[0, 1]$. From Theorem 1, $IP_\alpha = IR_\beta = 1$ only if $\mathbb{P}_g = \mathbb{P}_r$.

Together, $IP_\alpha$ and $IR_\beta$ serve as a measure of the discrepancy between the distributions $\mathbb{P}_r$ and $\mathbb{P}_g$, eliminating the need to augment our precision-recall analysis with measures of statistical divergence. Moreover, unlike $f$-divergence measures, the $(IP_\alpha, IR_\beta)$ metric disentangles fidelity and diversity into separate components, and does not require that $\mathbb{P}_r$ and $\mathbb{P}_g$ share a common support.

### 3.2.2 AUTHENTICITY

Generalization is independent of precision and recall since a model can achieve perfect fidelity and diversity without truly generating any samples, simply by resampling training data. Unlike discriminative models for which generalization is easily tested via held-out data, evaluating generalization in generative models is not straightforward (Adlam et al. (2019); Meehan et al. (2020)). We propose an *authenticity* score $A \in [0, 1]$ to quantify the rate by which a model generates *new* samples. To pin down a mathematical definition for $A$, we reformulate $\mathbb{P}_g$ as a mixture of densities as follows:

$$\mathbb{P}_g = A \cdot \mathbb{P}'_g + (1 - A) \cdot \delta_{g,\epsilon}, \tag{7}$$

where $\mathbb{P}'_g$ is the generative distribution conditioned on the synthetic samples being non-overfitted, and $\delta_{g,\epsilon}$ is a noisy distribution over training data. In particular, we define $\delta_{g,\epsilon}$ as $\delta_{g,\epsilon} = \delta_g * \mathcal{N}(0, \epsilon^2)$, where $\delta_g$ is a discrete distribution that places an unknown probability mass on each training data point in $\mathcal{D}_{real}$, $\epsilon$ is an arbitrarily small noise variance, and $*$ is the convolution operator. Essentially, (7) assumes that the model flips a (biased coin), pulling off a training sample with probability $1 - A$ and adding some noise to it, or innovating a new sample with probability $A$. A model with $A = 1$ always innovates, whereas an overfitted model will concentrate $\mathbb{P}_g$ around the training data.

## 4 ESTIMATING THE EVALUATION METRIC

Since the metrics in Section 3 are defined through binary conditions on individual samples, we can obtain an estimate $\widehat{\mathcal{E}} = (\widehat{P}_\alpha, \widehat{R}_\beta, \widehat{A})$ of the metric $\mathcal{E}$, for a given $\alpha$ and $\beta$, by assigning binary scores $\widehat{P}_{\alpha,j}, \widehat{A}_j \in \{0, 1\}$ to each synthetic sample $\widetilde{X}_{g,j}$ in $\mathcal{D}_{synth}$, and $\widehat{R}_{\beta,i} \in \{0, 1\}$ to each real sample $\widetilde{X}_{r,i}$ in $\mathcal{D}_{real}$, then averaging over all samples, i.e., $\widehat{P}_\alpha = \frac{1}{m} \sum_j \widehat{P}_{\alpha,j}$, $\widehat{R}_\beta = \frac{1}{n} \sum_i \widehat{R}_{\beta,i}$, $\widehat{A} = \frac{1}{m} \sum_j \widehat{A}_j$. To assign binary scores, we construct 3 classifiers $f_P, f_R, f_A : \widetilde{\mathcal{X}} \to \{0, 1\}$, where $\widehat{P}_{\alpha,j} = f_P(\widehat{X}_{g,j})$, $\widehat{R}_{\beta,i} = f_R(\widehat{X}_{r,i})$ and $\widehat{A}_j = f_A(\widehat{X}_{g,j})$. We explain the operation of each classifier in what follows.

**Precision and Recall classifiers.** Based on definitions (4) and (5), both classifiers check if a sample resides in an $\alpha$- (or $\beta$-) support, i.e., $f_P(\widetilde{X}_g) = \mathbf{1}\{\widetilde{X}_g \in \widehat{\mathcal{S}}_r^\alpha\}$ and $f_R(\widetilde{X}_r) = \mathbf{1}\{\widetilde{X}_r \in \widehat{\mathcal{S}}_g^\beta\}$. Hence, the main difficulty in implementing $f_P$ and $f_R$ is estimating the supports $\widehat{\mathcal{S}}_r^\alpha$ and $\widehat{\mathcal{S}}_g^\beta$—in fact, even if we know the exact distributions $\mathbb{P}_r$ and $\mathbb{P}_g$, computing their $\alpha$- and $\beta$-supports is not straightforward as it involves solving the optimization problem in (2).

To address this challenge, we pre-process the real and synthetic data in a way that renders estimation of $\alpha$-and $\beta$-supports straightforward. The idea is to train the evaluation embedding $\Phi$ so as to cast the supports of the real data, $\mathcal{S}_r$, into a *hypersphere* with radius $r$, and cast the distribution $\mathbb{P}_r$ into an isotropic density concentrated around the center $c_r$ of the hypersphere. We achieve this by modeling $\Phi$ as a *one-class* neural network trained with the following loss function: $L = \sum_i \ell_i$, where

$$\ell_i = r^2 + \frac{1}{\nu}\, \max\{0, \|\Phi(X_{r,i}) - c_r\|^2 - r^2\}. \tag{8}$$

The loss is minimized over the radius $r$ and the parameters of $\Phi$; the output dimensions of $\Phi$, $c_r$ and $\nu$ are viewed as hyperparameters (see Supplementary material). The loss in (8) is based on the seminal

work on one-class SVMs in (Schölkopf et al. (2001)), which is commonly applied to outlier detection problems, e.g., (Ruff et al. (2018)). In a nutshell, the evaluation embedding squeezes real data into the minimum-volume hypersphere centered around $c_r$, hence the real $\alpha$-support is estimated as:

$$\widehat{\mathcal{S}}_r^\alpha = \boldsymbol{B}(c_r, \widehat{r}_\alpha), \ \widehat{r}_\alpha = \widehat{Q}_\alpha\{\|\widetilde{X}_{r,i} - c_r\| : 1 \leq i \leq n\}, \tag{9}$$

where $\boldsymbol{B}(c, r)$ is a Euclidean ball with center $c$ and radius $r$, and $\widehat{Q}_\alpha$ is the $\alpha$-quantile function. The set of all $\alpha$-supports of $\mathbb{P}_r$ corresponds to the set of all concentric spheres with center $c_r$ and radii $\widehat{r}_\alpha, \forall \alpha \in [0, 1]$. Thus, the precision classifier assigns a score 1 to a synthetic sample $\widetilde{X}_g$ if it resides in the Ball $\widehat{\mathcal{S}}_r^\alpha$, i.e., $f_p(\widetilde{X}_g) = \mathbf{1}\{\|\widetilde{X}_g - c_r\| \leq \widehat{r}_\alpha\}$. Now define $c_g = \frac{1}{m} \sum_j \widetilde{X}_{g,j}$, and consider a hypersphere $\boldsymbol{B}(c_g, \widehat{r}_\beta)$, where $\widehat{r}_\beta = \widehat{Q}_\beta\{\|\widetilde{X}_{g,j} - c_g\| : 1 \leq j \leq m\}$. We construct $f_R$ as follows:

$$f_R(\widetilde{X}_{r,i}) = \mathbf{1}\{\widetilde{X}_{g,j^*}^\beta \in \boldsymbol{B}(\widetilde{X}_{r,i}, \text{NND}_k(\widetilde{X}_{r,i}))\}, \tag{10}$$

where $\widetilde{X}_{g,j^*}^\beta$ is the synthetic sample in $\boldsymbol{B}(c_g, \widehat{r}_\beta)$ that is closest to $\widetilde{X}_{r,i}$, and $\text{NND}_k(\widetilde{X}_{r,i})$ is the distance between $\widetilde{X}_{r,i}$ and its $k$-nearest neighbor in $\mathcal{D}_{real}$. Similar to the estimator in (Naeem et al. (2020)), (10) is a nonparametric estimate of $\mathcal{S}_g^\beta$ that checks if each real sample $i$ is locally covered by a synthetic sample in $\boldsymbol{B}(c_g, \widehat{r}_\beta)$. A discussion on how to select the hyper-parameter $k$, as well as an alternative method for estimating $\mathcal{S}_g^\beta$ using one-class representations is provided in the Appendix.

**Authenticity classifier.** We derive the classifier $f_A$ from a hypothesis test that tests if a sample $\widetilde{X}_{g,j}$ is non-memorized. Let $\mathcal{H}_1 : A_j = 1$ be the hypothesis that $\widetilde{X}_{g,j}$ is authentic, with the null hypothesis $\mathcal{H}_0 : A_j = 0$. To test the hypothesis, we use the likelihood-ratio statistic (Van Trees (2004)):

$$\Lambda(\widetilde{X}_{g,j}) = \mathbb{P}(\widetilde{X}_{g,j} \,|\, A_j = 1)/\mathbb{P}(\widetilde{X}_{g,j} \,|\, A_j = 0) = \mathbb{P}'_g(\widetilde{X}_{g,j})/\delta_{g,\epsilon}(\widetilde{X}_{g,j}), \tag{11}$$

which follows from (7). Since both likelihood functions in (11) are unknown, we need to test the hypothesis $\mathcal{H}_1 : A_j = 1$ using an alternative sufficient statistic with a known probability distribution.

Let $d_{g,j} = d(\widetilde{X}_{g,j}, \mathcal{D}_{real})$ be the distance between synthetic sample $j$ and the training data set, and let $i^*$ be the training sample in $\mathcal{D}_{real}$ closest to $X_{g,j}$, i.e., $d_{g,j} = d(\widetilde{X}_{g,j}, \widetilde{X}_{r,i^*})$. Let $d_{r,i^*}$ be the distance between $\widetilde{X}_{r,i^*}$ and $\mathcal{D}_{real}/\{\widetilde{X}_{r,i^*}\}$, i.e., the training data with sample $i^*$ removed. Now consider the statistic $a_j = \mathbf{1}\{d_{g,j} \leq d_{r,i^*}\}$, which indicates if synthetic sample $j$ is closer to training data than any other training sample. The likelihood ratio for observations $\{a_j\}_j$ under hypotheses $\mathcal{H}_0$ and $\mathcal{H}_1$ is

$$\Lambda(a_j) = \mathbb{P}(a_j \,|\, A_j = 1)/\mathbb{P}(a_j \,|\, A_j = 0) \approx a_j^{-1} \cdot \mathbb{P}(d_{g,j} \leq d_{r,i^*} \,|\, A_j = 1). \tag{12}$$

Here, we used the fact that if sample $j$ is a memorized copy of $i^*$, and if the noise variance $\epsilon$ in (7) is arbitrarily small, then $a_j = 1$ almost surely and $\mathbb{P}(a_j \,|\, A_j = 0) \approx 1$. If $j$ is authentic, then $\widetilde{X}_{g,j}$ lies in the convex hull of the training data, and hence $\mathbb{P}(a_j \,|\, A_j = 0) \to 0$ and $\Lambda \to \infty$ for a large real data set. Thus, $f_A$ issues a label $A_j = 1$ if $a_j = 0$, and $A_j = 0$ otherwise. Intuitively, $f_A$ deems sample $j$ unauthentic if it is closer to $i^*$ than any other real sample in the training data.

## 5 EXPERIMENTS AND USE CASES

### 5.1 EVALUATING & AUDITING GENERATIVE MODELS FOR SYNTHESIZING COVID-19 DATA

In this experiment, we use our metric to assess the ability of different generative models to synthesize COVID-19 patient data that can be used for predictive modeling. Using SIVEP-Gripe (SIVEP-Gripe (2020)), a database of 99,557 COVID patients in Brazil, including sensitive data such as ethnicity. We use generative models to synthesize replicas of this data and fit predictive models to the replicas.

**Models and baselines.** We create 4 synthetic data sets using GAN, VAE, Wasserstein GANs with a gradient penalty (WGAN-GP) (Gulrajani et al. (2017)), and ADS-GAN, which is specifically designed to prevent patient identifiablity in generated data (Yoon et al. (2020)). To evaluate these synthetic data sets, we use Fréchet Inception Distance (FID) (Heusel et al. (2017)), Precision/Recall ($P_1/R_1$) (Sajjadi et al. (2018)), Density/Coverage ($D/C$) (Naeem et al. (2020)), Parzen window likelihood ($PW$) (Bengio et al. (2013)) and Wasserstein distance ($W$) as baselines. On each synthetic data, we fit a predictive Logistic regression model to predict patient-level COVID-19 mortality.

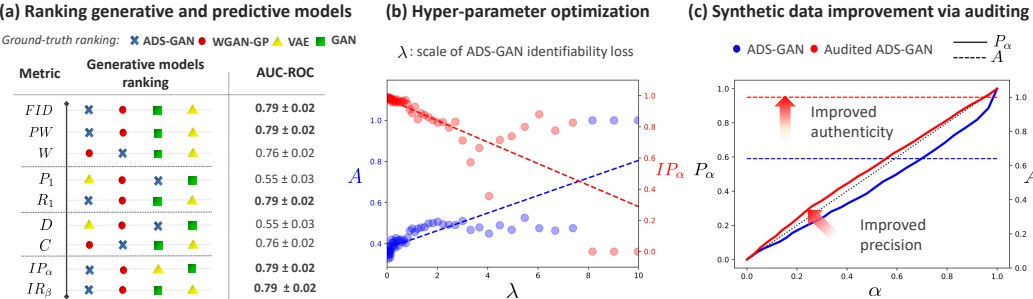

Figure 4: **Predictive modeling with synthetic data. (a)** Here, we rank the 4 generative models (ADS-GAN: $\times$, WGAN-GP: $\bullet$, VAE: $\blacktriangle$, GAN: $\blacksquare$) with respect to each evaluation metric (leftmost is best). For each metric, we train a predictive model on the synthetic data set with highest score, and test its AUC on real data. Ground-truth ranking of synthetic data is the ranking of the AUC of predictive models trained on them. (b) Hyper-parameter tuning for ADS-GAN. (Dashed lines are linear regression lines.) (c) Post-hoc auditing of ADS-GAN.

**Predictive modeling with synthetic data.** In the context of predictive modeling, a generative model is assessed with respect to its usefulness in training predictive models that generalize well on real data. Hence, the "ground-truth" ranking of the 4 generative models corresponds to the ranking of the AUC-ROC scores achieved by predictive models fit to their respective synthetic data sets and tested on real data (Figure 4(a)). The data synthesized by ADS-GAN ($\times$) displayed the best performance, followed by WGAN-GP ($\bullet$), VAE ($\blacktriangle$), and GAN ($\blacksquare$). To assess the accuracy of baseline evaluation metrics, we test if each metric can recover the ground-truth ranking of the 4 generative models (Figure 4(a)). Our integrated precision and recall metrics $IP_\alpha$ and $IR_\beta$ both assign the highest scores to ADS-GAN; $IP_\alpha$ exactly nails the right ranking of generative models. On the other hand, competing metrics such as $P_1$, $C$ and $D$, over-estimate the quality of VAE and WGAN-GP—if we use these metrics to decide which generative model to use, we will end up with predictive models that perform poorly, i.e. AUC-ROC of the predictive model fitted to synthetic data with best $P_1$ is 0.55, compared to an AUC-ROC of 0.79 for our $IP_\alpha$ score.

These results highlight the importance of accounting for the densities $\mathbb{P}_g$ and $\mathbb{P}_r$, and not just their supports, when evaluating a generative model. This is because a shifted $\mathbb{P}_g$ would result in a "covariate shift" in synthetic data, leading to poor generalization for predictive models fitted to it, even when real and synthetic support coincide. As we can see in Figure 4(a), metrics that compare distributions (our metrics, $PW$ and $FID$), are able to accurately rank the 4 generative models.

**Hyper-parameter tuning & the privacy-utility tradeoff.** Another use case for our metric is hyper-parameter optimization for generative models. Here, we focus on the best-performing model in our previous experiment: ADS-GAN. This model has a hyper-parameter $\lambda \in \mathbb{R}$ that determines the importance of the privacy-preservation loss function used to regularize the training of ADS-GAN (Yoon et al. (2020)): smaller values of $\lambda$ make the model more prone to overfitting, and hence privacy leakage. Figure 4(b) shows how our precision and authenticity metrics change with the different values of $\lambda$: the curve provides an interpretable tradeoff between privacy and utility (e.g., for $\lambda = 2$, an $A$ score of 0.4 means that 60% of patients may have personal information exposed). Increasing $\lambda$ improves privacy at the expense of precision. By visualizing this tradeoff using our metric, data holders can understand the risks of different modeling choices involved in data synthesis.

**Improving synthetic data via model auditing.** Our metrics are not only useful for hyper-parameter tuning, but can also be used to improve the quality of synthetic data generated by an already-trained model using (post-hoc) auditing. Because our metrics are defined on the sample level, we can discard unauthentic or imprecise samples. This does not only lead to nearly optimal precision and authenticity for the curated data (Figure 4(c)), but also improves the AUC-ROC of the predictive model fitted to audited data (from 0.76 to 0.78 for the audited ADS-GAN synthetic data, $p < 0.005$), since auditing eliminates noisy data points that would otherwise undermine generalization performance.

## 5.2 DIAGNOSING GENERATIVE DISTRIBUTIONS OF MNIST

In this experiment, we test the ability of our metrics to detect common modes of failure in generative modeling—in particular, we emulate a *mode dropping* scenario, where the generative model fails to

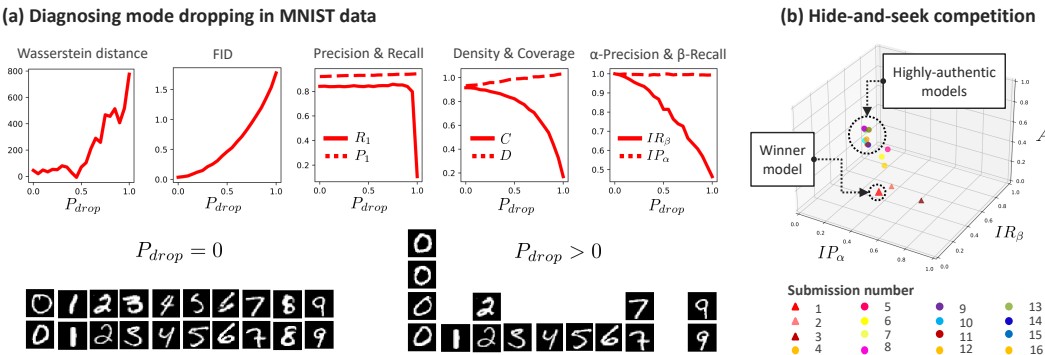

Figure 5: (a) Diagnosing mode collapse in MNIST data. (b) Results for the hide-and-seek competition.

recognize the distinct modes in a multimodal distribution $\mathbb{P}_r$, and instead recovers a single mode in $\mathbb{P}_g$. To construct this scenario, we fit a conditional GAN (CGAN) model (Wang et al. (2018)) on the MNIST data set, and generate 1,000 samples for each of the digits 0-9. (We can think of each digit as a distinct mode in $\mathbb{P}_r$.) To apply mode dropping, we first sample 1,000 instances of each digit from the CGAN, and then delete individual samples of digits 1 to 9 with a probability $P_{drop}$, and replace the deleted samples with new samples of the digit 0 to complete a data set of 10,000 instances. The parameter $P_{drop} \in [0, 1]$ determines the severity of mode dropping: for $P_{drop} = 0$, the data set has all digits being equally represented with 1,000 samples, and for $P_{drop} = 1$, the data set has 10,000 samples of the digit 0 only as depicted pictorially in Figure 5(a) (bottom panel).

We show how the different evaluation metrics respond to varying $P_{drop}$ from 0 to 1 in Figure 5(a) (top). Because mode dropping pushes the generative distribution away from the real one, statistical distance metrics such as $W$ and $FID$ increase as $P_{drop}$ approaches 1. However, these metrics only reflect a discrepancy between $\mathbb{P}_r$ and $\mathbb{P}_g$, and do not disentangle the Fidelity and Diversity components of this discrepancy. On the other hand, standard precision and recall metric are completely insensitive to mode dropping except for the extreme case when $P_{drop} = 1$. This is because both metrics only check supports of $\mathbb{P}_r$ and $\mathbb{P}_g$, so they cannot recognize mode dropping as long as there is a non-zero probability that the model will generates digits 1-9. On the contrary, mode dropping reflects in our metrics, which manifest in a declining $IR_\beta$ as $P_{drop}$ increases. Since mode dropping affects coverage of digits and not the quality of images, it only affects $IR_\beta$ but not $IP_\alpha$.

## 5.3 REVISITING THE HIDE-AND-SEEK CHALLENGE FOR SYNTHESIZING TIME-SERIES DATA

Finally, we use our metric to re-evaluate the generative models submitted to the NeurIPS 2020 Hide-and-Seek competition (Jordon et al. (2020)). In this competition, participants were required to synthesize intensive care time-series data based on real data from the AmsterdamUMCdb database. A total of 16 submissions were judged based on the accuracy of predictive models fit to the synthetic data (an approach similar to the one in Section 5.1). The submissions followed various modeling choices, including recurrent GANs, autoencoders, differential privacy GANs, etc. Details of all submissions are available online. Surprisingly, the winning submission was a very simplistic model that adds Gaussian noise to the real data to create new samples.

To evaluate our metrics on time-series data, we trained a Seq-2-Seq embedding that is augmented with our One-class representations to transform time-series into fixed feature vectors. (The architecture for this embedding is provided in the Supplementary material.) In Figure 5(b), we evaluate all submissions with respect to precision, recall and authenticity. As we can see, the winning submission comes out as one of the least authentic models, despite performing competitively in terms of precision and recall. This highlights the detrimental impact of using naïve metrics for evaluating generative models—based on the competition results, clinical institutions seeking to create synthetic data sets may be led to believe that Submission 1 in Figure 5(b) is the right model to use. However, our metrics—which give a fuller picture of the true quality of all submissions—shows that such model creates unauthentic samples that are mere noisy copies of real data, which would pose risk to patient privacy. We hope that our metrics and our pre-trained Seq-2-Seq embeddings can help clinical institutions evaluate the quality of their synthetic time-series data in the future.

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

SUPPLEMENTARY MATERIAL

APPENDIX A: LITERATURE REVIEW

In this Section, we provide a comprehensive survey of prior work, along with a detailed discussion on how our metric relates to existing ones. We classify existing metrics for evaluating generative models into two main classes:

1. **Statistical divergence metrics**

2. **Precision and recall metrics**

Divergence metrics are single-valued measures of the distance between the real and generative distributions, whereas precision-recall metrics classify real and generated samples as to whether they are covered by generative and real distributions, respectively. In what follows, we list examples of these two types of metrics, highlighting their limitations.

**Statistical divergence metrics.**

The most straightforward approach for evaluating a generative distribution is to compute the model log-likelihood—for density estimation tasks, this has been the de-facto standard for training and evaluating generative models. However, the likelihood function is a model-dependent criteria: this is problematic because the likelihood of many state-of-the-art models is inaccessible. For instance, GANs are implicit likelihood models and hence provide no explicit expression for its achieved log-likelihood Goodfellow et al. (2014). Other models, energy-based models has a normalization constant in the likelihood expression that is generally difficult to compute as they require solving intractable complex integrals Kingma & Welling (2013).

Statistical divergence measures are alternative (model-independent) metrics that are related to log-likelihood, and are commonly used for training and evaluating generative models. Examples include lower bounds on the log-likelihood Kingma & Welling (2013), contrastive divergence and noise contrastive estimation Hinton (2002); Gutmann & Hyvärinen (2010), probability flow Sohl-Dickstein et al. (2011), score matching Hyvärinen et al. (2009), maximum mean discrepancy (MMD) Gretton et al. (2012), and the Jensen-Shannon divergence (JSD).

In general, statistical divergence measures suffer from the following limitations. The first limitation is that likelihood-based measures can be inadequate in high-dimensional feature spaces. As has been shown in (Theis et al., 2015), one can construct scenarios with poor likelihood and great samples through a simple lookup table model, and vice versa, we can think of scenarios with great likelihood and poor samples. This is because, if the model samples white noise 99% of the time, and samples high-quality outputs 1% of the time, the log-likelihood will be hardly distinguishable from a model that samples high-quality outputs 100% of the time if the data dimension is large. Our metrics solve this problem by measuring the rate of error on a sample-level rather than evaluating the overall distribution of samples.

Moreover, statistical divergence measures collapse the different modes of failure of the generative distribution into a single number. This hinders our ability to diagnose the different modes of generative model failures such as mode dropping, mode collapse, poor coverage, etc.

**Precision and recall metrics.**

Precision and recall metrics for evaluating generative models were originally proposed in Sajjadi et al. (2018). Our metrics differ from these metrics in various ways. First, unlike standard metrics, $\alpha$-Precision and $\beta$-Recall take into account not only the supports of $\mathbb{P}_r$ and $\mathbb{P}_g$, but also the actual probability densities of both distributions. Standard precision (and recall) correspond to one point on the $P_\alpha$ (and $R_\beta$) curve; they are equal to $P_\alpha$ and $R_\beta$ evaluated on the full support (i.e., $P_1$ and $R_1$). By defining our metrics with respect to the $\alpha$- and $\beta$-supports, we do not treat all samples equally, but rather assign higher importance to samples that land in "denser" regions of $\mathcal{S}_r$ and $\mathcal{S}_g$. Hence, $P_\alpha$ and $R_\beta$ reflect the extent to which $\mathbb{P}_r$ and $\mathbb{P}_g$ are *calibrated*—i.e., good $P_\alpha$ and $R_\beta$ curves are achieved when $\mathbb{P}_r$ and $\mathbb{P}_g$ share the same modes and not just a common support. While optimal $R_1$ and $P_1$ can be achieved by arbitrarily mismatched $\mathbb{P}_r$ and $\mathbb{P}_g$, our $P_\alpha$ and $R_\beta$ curves are optimized only when $\mathbb{P}_r$ and $\mathbb{P}_g$ are identical as stated by Theorem 1.

The new $P_\alpha$ and $R_\beta$ metrics address the major shortcomings of precision and recall. Among these shortcomings are: lack of robustness to outliers, failure to detect matching distributions, and inability to diagnose different types of distributional failure (such as mode collapse, mode invention, or density shifts) Naeem et al. (2020). Basically, a model $\mathbb{P}_g$ will score perfectly on precision and recall ($R_1 = P_1 = 1$) as long as it nails the support of $\mathbb{P}_r$, even if $\mathbb{P}_r$ and $\mathbb{P}_g$ place totally different densities on their common support.

In addition to the above, our metrics estimate the supports of real and generative distributions using neural networks rather than nearest neighbor estimates as in Naeem et al. (2020). This prevents our estimates from overestimating the supports of real and generative distributions, thereby overestimating the coverage or quality of the generated samples.

## APPENDIX B: PROOF OF THEOREM 1

To prove the statement of the Theorem, we need to prove the two following statements:

(1) $\mathbb{P}_g = \mathbb{P}_r \rightarrow P_\alpha/\alpha = R_\beta/\beta = 1, \forall \alpha, \beta$

(2) $P_\alpha/\alpha = R_\beta/\beta = 1, \forall \alpha, \beta \rightarrow \mathbb{P}_g = \mathbb{P}_r$

To prove (1), we start by noting that since we have $\mathbb{P}_g = \mathbb{P}_r$, then $\mathcal{S}_\alpha^g = \mathcal{S}_\alpha^r, \forall \alpha \in [0, 1]$. Thus, we have

$$P_\alpha = \mathbb{P}(\widetilde{X}_g \in \mathcal{S}_r^\alpha) = \mathbb{P}(\widetilde{X}_g \in \mathcal{S}_g^\alpha) = \alpha, \tag{13}$$

for all $\alpha \in [0, 1]$, and similarly, we have

$$R_\beta = \mathbb{P}(\widetilde{X}_r \in \mathcal{S}_g^\beta) = \mathbb{P}(\widetilde{X}_r \in \mathcal{S}_r^\beta) = \beta, \tag{14}$$

for all $\beta \in [0, 1]$, which concludes condition (1).

Now we consider condition (2). We first note that $\mathcal{S}_r^\alpha \subseteq \mathcal{S}_r^{\alpha'}$ for all $\alpha' > \alpha$. If $P_\alpha = \alpha$ for all $\alpha$, then we have

$$\mathbb{P}(\widetilde{X}_g \in \mathcal{S}_r^\alpha) = \int_{\mathcal{S}_r^\alpha} d\mathbb{P}_g = \alpha, \forall \alpha \in [0, 1]. \tag{15}$$

Now assume that $\alpha' = \alpha + \Delta\alpha$, then we have

$$\int_{\mathcal{S}_r^{\alpha'}/\mathcal{S}_r^\alpha} d\mathbb{P}_g = \int_{\mathcal{S}_r^{\alpha'}/\mathcal{S}_r^\alpha} d\mathbb{P}_r = \Delta\alpha. \tag{16}$$

Thus, the probability masses of $\mathbb{P}_g$ and $\mathbb{P}_r$ are equal for all infinitesimally small region $\mathcal{S}_r^{\alpha+\Delta\alpha}/\mathcal{S}_r^\alpha$ (for $\Delta\alpha \to 0$) of the $\alpha$-support of $\mathbb{P}_r$, hence $\mathbb{P}_g = \mathbb{P}_r$ for all subsets of $\mathcal{S}_r^1$. By applying the similar argument to the recall metric, we also have $\mathbb{P}_g = \mathbb{P}_r$ for all subsets of $\mathcal{S}_g^1$, and hence $\mathbb{P}_g = \mathbb{P}_r$.

## APPENDIX C: ALTERNATIVE APPROACH FOR ESTIMATING THE SUPPORT OF SYNTHETIC DATA & CODE SNIPPETS

Instead of using a $k$-NN approach to estimate the generative support $\mathcal{S}_g^\beta$, one could use a separate one-class representation $\Phi_g$ for each new synthetic sample being evaluated. We provide code snippets and comparisons between the two approaches in the an anonymized Colab notebook. While the two approaches perform rather similarly, we opt to adopt the $k$-NN based approach to avoid potential biases induced by using a separate representation for each generative model when using our metric for model comparisons.

## APPENDIX D: EXPERIMENTAL DETAILS

### .1 DATA

In this research the argue for the versatility of our metrics, hence we have included results for tabular (static), time-series and image data (see Table 1). For the tabular data we use Baqui et al. (2020)'s

Table 1: Datasets used, with $n$ and $d$ the number of samples and features, respectively.

| Name | Type | $n$ | $d$ | Embedding | $d_{emb}$ |
|---|---|---|---|---|---|
| SIVEP-GRIPE | Tabular | 6882 | 25 | - | - |
| AmsterdamUMCdb | Time-series | 7695 | 70 | Seq-2-Seq | 280 |
| MNIST | Image | 10000 | 784 | InceptionV3 | 2048 |

preprocessed version of the SIVEP-GRIPE dataset of Brazilian ICU Covid-19 patient data. For the image experiments, we use the 10,000 samples in the default MNIST test set LeCun (1998). For proper evaluation of the authenticity metric, the same original data is used for training of generative models and evaluation of all metrics.

For the time-series experiments, AmsterdamUMCdb is used in a manner exactly analogous to the NeurIPS 2020 Hide-and-Seek Privacy Challenge Jordon et al. (2020), which describes it as follows: "AmsterdamUMCdb was developed and released by Amsterdam UMC in the Netherlands and the European Society of Intensive Care Medicine (ESICM). It is the first freely accessible comprehensive and high resolution European intensive care database. It is also first to have addressed compliance with General Data Protection Regulation [...] AmsterdamUMCdb contains approximately 1 billion clinical data points related to 23,106 admissions of 20,109 unique patients between 2003 and 2016. The released data points include patient monitor and life support device data, laboratory measurements, clinical observations and scores, medical procedures and tasks, medication, fluid balance, diagnosis groups and clinical patient outcomes.". Notably, only the longitudinal features from this database are kept, with static ones discarded. The same subset as was used in the competition for "hider" synthetic data generation is used; this consists of 7695 examples with 70 features (and a time column), sequence length is limited to 100 (the original data contains sequences of up to length 135,337). The features are normalised to $[0, 1]$ and imputed as follows: (1) back-fill, (2) forward-fill, (3) feature median imputation. This preprocessing is chosen to match the competition Jordon et al. (2020). The competition "hider" submissions were trained on this dataset and the synthetic data generated.

For metric consistency and the avoidance of tedious architecture optimization for each data modality, we follow previous works (e.g. Heusel et al. (2017); Sajjadi et al. (2018); Kynkäänniemi et al. (2019); Naeem et al. (2020)) and embed image and time series data into a static embedding. This is required, since the original space is non-euclidean and will result in failure of most metrics. The static embedding is used for computing baseline metrics, and is used as input for the One-Class embedder.

For finding static representations of MNIST, images are upscaled and embedded using InceptionV3 pre-trained on ImageNET without top layer. This is the same embedder used for computing Frchet Inception Distance Heusel et al. (2017). Very similar results were obtained using instead a VGG-16 embedder Brock et al. (2018); Kynkäänniemi et al. (2019). Preliminary experimentation with random VGG-16 models Naeem et al. (2020) did not yield stable results for neither baselines nor our methods.

### .2 TIME SERIES EMBEDDING

The time series embeddings used throughout this work are based on *Unsupervised Learning of Video Representations using LSTMs* Srivastava et al. (2015), specifically the "LSTM Autoencoder Mode". A sequence-to-sequence LSTM network is trained, with the target sequence set as the input sequence (reversed for ease of optimization), see Figure 1. The encoder hidden and cell states ($h$ and $c$ vectors) at the end of a sequence are used as the learned representation and are passed to the decoder during training. At inference, these are concatenated to obtain one fixed-length vector per example.

The specifics of the LSTM autoencoder used here are as follows. Two LSTM layers are used in each encoder and decoder. The size of $h, c$ vectors is 70 (280 after concatenation). The model was implemented in PyTorch Paszke et al. (2017), utilising sequence packing for computational efficiency. All autoencoders were trained to convergence on the original data; the synthetic time series data was passed through this at inference. The time column (when present in data) was discarded.

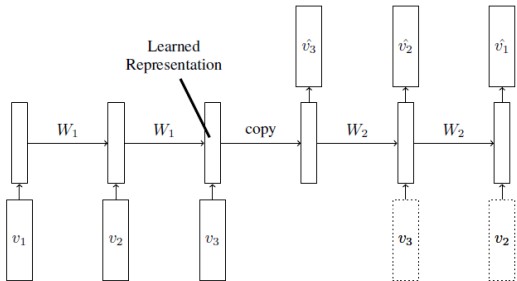

Figure 1: Architecture of LSTM Autoencoder, sourced from Srivastava et al. (2015).

Table 2: Metrics on tabular data for different generative models. Row "audited" contains results for data generated by ADS-GAN, but in which samples are rejected if they do not meet the precision or authenticity threshold.

| Model | W | FD | Parzen | $P_1$ | $R_1$ | $D$ | $C$ | $A$ | $IP_\alpha$ | $IR_\beta$ |
|---|---|---|---|---|---|---|---|---|---|---|
| VAE | 2.2626 | 2.9697 | -11.3653 | 1.0000 | 0.1022 | 1.3999 | 0.0926 | 0.5147 | 0.4774 | 0.0596 |
| GAN | 1.5790 | 1.8065 | -10.7824 | 0.7059 | 0.0875 | 0.4316 | 0.0939 | 0.6077 | 0.9802 | 0.0648 |
| WGAN-GP | -0.0194 | 0.0856 | -8.3650 | 0.9439 | 0.7299 | 1.0709 | 0.8945 | 0.4712 | 0.9398 | 0.4468 |
| ADS-GAN | 0.3578 | 0.2134 | -8.7952 | 0.8083 | 0.6133 | 0.4711 | 0.5357 | 0.5905 | 0.7744 | 0.2914 |
| DPGAN | 1.1216 | 0.9389 | -8.8394 | 0.9923 | 0.1822 | 1.4885 | 0.5065 | 0.3793 | 0.9591 | 0.1863 |
| audited | -0.0470 | 0.0600 | -8.5408 | 0.8986 | 0.8737 | 0.7560 | 0.8050 | 1.0000 | 0.9994 | 0.1961 |

## .3 FULL RESULTS

Table 2 contains metrics computed on different generated versions of the SIVEP-GRIPE tabular dataset. Included metrics are Wasserstein distance, Frchet Distance ($FD$), Parzen window likelihood estimate, precision $P_1$, recall $R_1$, density ($D$), coverage $C$ and the proposed metrics, specifically integrated $\alpha$-precision $IP_\alpha$, integrated $\beta$-recall $IR_\beta$ and authenticity $A$. For the tabular data, data is generated using a VAE, GAN, Wasserstein GAN with gradient penalisation (WGAN-GP) Arjovsky et al. (2017), ADS-GAN Yoon et al. (2020), Differentially Private GAN (DP-GAN) Xie et al. (2018) and an ADS-GAN generated dataset in which samples are audited on precision and authenticity. Similarly, Table 3 contains metric results[1] for MNIST, generated by a VAE, Deep convolution GAN (DCGAN), WGAN-GP and ADS-GAN. Table 4 contains results for MIMIC generation using different methods from the Hide-and-Seek Privacy Competition Jordon et al. (2020). The submission that won the competition is the penultimate model, Hamada. The last row shows results for an audited version of the Hamada dataset, in which we keep generating data using the Hamada model and discard samples that do not meet the precision or authenticity threshold.

---

[1]Note that here, the Frchet Distance equals the Frchet Inception Distance, because the metrics are computed on InceptionV3 embeddings of the image data.

Table 3: Metrics on MNIST data for different generative models.

| Model | W | FD | Parzen | $P_1$ | $R_1$ | $D$ | $C$ | $A$ | $IP_\alpha$ | $IR_\beta$ |
|---|---|---|---|---|---|---|---|---|---|---|
| VAE | 606.5 | 112934 | -349913 | 0.2160 | 0.0140 | 0.0885 | 0.0810 | 0.8167 | 0.4280 | 0.1452 |
| DCGAN | -98.5 | 2578 | -180132 | 0.8947 | 0.8785 | 0.8589 | 0.9071 | 0.6059 | 0.9889 | 0.4815 |
| WGAN-GP | -64.8 | 4910 | -185745 | 0.8931 | 0.8504 | 0.8084 | 0.8509 | 0.6146 | 0.9894 | 0.4199 |
| ADS-GAN | -114.1 | 574 | -28657 | 1.0000 | 0.9998 | 1.1231 | 1.0000 | 0.5268 | 0.9900 | 0.5549 |

Table 4: Metrics on AmsterdamUMCdb data for different generative models.

| Model (CodaLab username) | W | FD | Parzen | $P_1$ | $R_1$ | $D$ | $C$ | $A$ | $IP_\alpha$ | $IR_\beta$ |
|---|---|---|---|---|---|---|---|---|---|---|
| Add noise [baseline] | 40.6 | 497 | -228 | 0.0013 | 0.0173 | 0.0003 | 0.0010 | 0.8450 | 0.5187 | 0.0148 |
| Time-GAN [baseline] | 249.0 | 30478 | -1404 | 0.0000 | 0.6642 | 0.0000 | 0.0000 | 0.9991 | 0.5000 | 0.0005 |
| akashdeepsingh | 13.3 | 44 | -34 | 0.7643 | 0.3566 | 0.4263 | 0.0741 | 0.3958 | 0.6074 | 0.1408 |
| saeedsa | 93.4 | 1509 | -2597 | 0.0000 | 0.0000 | 0.0000 | 0.0000 | 0.9925 | 0.5018 | 0.0025 |
| wangzq312 | 62.3 | 645 | -119 | 0.4430 | 0.8630 | 0.0920 | 0.0061 | 0.7665 | 0.6116 | 0.0185 |
| csetraynor | 64.6 | 1604 | -4710 | 0.0000 | 0.0000 | 0.0000 | 0.0000 | 1.0000 | 0.5339 | 0.0000 |
| jilljenn | 118.0 | 17235 | -5874 | 0.0000 | 0.0000 | 0.0000 | 0.0000 | 1.0000 | 0.4999 | 0.0000 |
| SatoshiHasegawa | 2.4 | 33 | -156 | 0.9333 | 0.3264 | 0.5626 | 0.0988 | 0.3172 | 0.8050 | 0.1717 |
| flynngo | 126.3 | 8786 | -120 | 0.0359 | 0.7663 | 0.0087 | 0.0022 | 0.8448 | 0.6497 | 0.0271 |
| tuscan-chicken-wrap | 118.7 | 5344 | -892 | 0.0000 | 0.0535 | 0.0000 | 0.0000 | 0.9854 | 0.5030 | 0.0053 |
| Atrin | 175.3 | 15159 | -1933 | 0.0000 | 0.7306 | 0.0000 | 0.0000 | 0.9975 | 0.5008 | 0.0019 |
| wangz10 | 113.8 | 4624 | -221 | 0.0039 | 0.2993 | 0.0010 | 0.0019 | 0.8509 | 0.5141 | 0.0113 |
| yingruiz | 146.7 | 8973 | -562 | 0.0069 | 0.0626 | 0.0014 | 0.0001 | 0.8995 | 0.5129 | 0.0062 |
| yingjialin | 138.9 | 8919 | -570 | 0.0000 | 0.0613 | 0.0000 | 0.0000 | 0.8864 | 0.5011 | 0.0030 |
| lumip | 25.8 | 271 | -78 | 0.1796 | 0.4749 | 0.0727 | 0.0256 | 0.6335 | 0.5938 | 0.0801 |
| hamada | 8.3 | 73 | -192 | 0.8933 | 0.4010 | 0.5906 | 0.0398 | 0.3572 | 0.5482 | 0.0850 |
| hamada [audited] | 7.1 | 48 | -204 | 0.9006 | 0.0665 | 0.5931 | 0.0444 | 1.0 | 0.9976 | 0.0334 |

## .4 HYPERPARAMETER OPTIMIZATION

### .4.1 BASELINES

For computing the density and coverage metrics, we set a threshold of 0.95 on the minimum expected coverage, as recommended in the original work (Eq. 9 Naeem et al. (2020)). For all datasets, this is achieved for $k = 5$. For consistency in these comparisons, we use $k = 5$ for the precision and recall metrics too.

### .4.2 ONECLASS EMBEDDINGS

We use Deep SVDD Ruff et al. (2018) to embed static data into One-Class representations. To mitigate hypersphere collapse (Propostions 2 and 3 of Ruff et al. (2018)), we do not include a bias term and use ReLU activation for the One-Class embedder. Original data is split into training (80%) and validation (20%) set, and One-Class design is fine-tuned to minimise validation loss. We use the SoftBoundary objective (Eq. 3 Ruff et al. (2018)) with $\nu = 0.01$ and center $\mathbf{c} = \mathbf{1}$ for tabular and time-series data and $\mathbf{c} = 10 \cdot \mathbf{1}$ for image data. Let $n_h$ be the number of hidden layers with each $d_h$ nodes, and let $d_z$ be the dimension of the representation layer. For tabular data, we use $n_h = 3$, $d_h = 32$ and $d_z = 25$; for time-series data, $n_h = 2$, $d_h = 128$ and $d_z = 32$; and for MNIST $n_h = 3$, $d_h = 128$ and $d_z = 32$. Models are implemented in PyTorch Paszke et al. (2017) and the AdamW optimizer is used with weight decay $10^{-2}$.

For the $\beta$-recall metric, estimating the support of synthetic data involves tuning the $k$ parameter of the $k$-NN estimator. The $k$ parameter can be tuned by fitting the NN estimator on a portion of the data for every given $k$, and then testing the recall on a held out (real) sample. The selected $k$ for each $\alpha$ is the smallest $k$ that covers $\alpha$ held out samples. Similar to Naeem et al. (2020), we found that through this procedure, $k = 5$ seems to come up as the optimal $k$ for most experiments.

## .5 TOY EXPERIMENTS

We include two toy experiments that highlight the advantage of the proposed metrics compared to previous works. We focus our comparison on the improved precision and recall Kynkäänniemi et al. (2019) and density and coverage Naeem et al. (2020) metrics.

### .5.1 ROBUSTNESS TO OUTLIERS

Naeem et al. (2020) showed that the precision and recall metrics as proposed by Sajjadi et al. (2018); Kynkäänniemi et al. (2019) are not robust to outliers. We replicate toy experiments to show the proposed $\alpha$-Precision and $\beta$-Recall do not suffer the same fate.

Let $X, Y \in \mathbb{R}^d$ denote original and synthetic samples respectively, with original $X \sim N(0, I)$ and $Y \sim N(\mu, I)$. We compute all metrics for $\mu \in [-1, 1]$. In this setting we conduct three experiments:

1. No outliers
2. One outlier in the real data at $X = \mathbf{1}$
3. One outlier in the synthetic data at $Y = \mathbf{1}$

We set $d = 64$ and both original and synthetic data we sample $10000$ points. Subsequent metric scores are shown in Figure

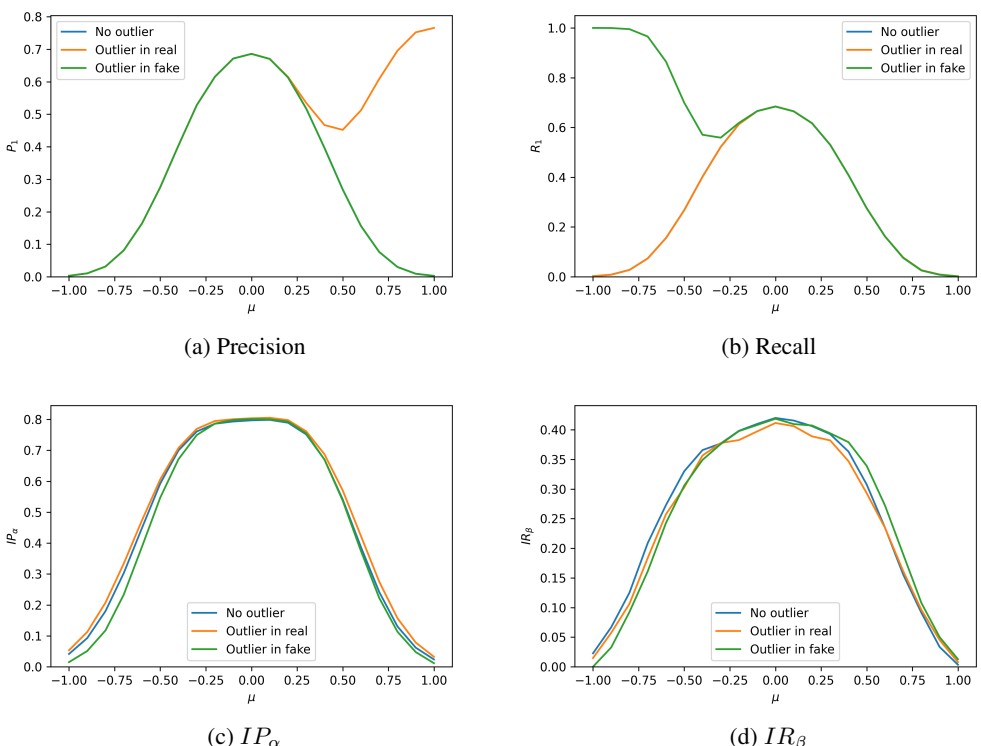

(a) Precision

(b) Recall

(c) $IP_\alpha$

(d) $IR_\beta$

Figure 2: Toy experiment I: outlier robustnes

As can be seen, the precision and recall metrics are not robust to outliers, as just a single outlier has dramatic effects. The $IP_\alpha$ and $IR_\beta$ are not affected, as the outlier does not belong to the $\alpha$-support (or $\beta$-support) unless $\alpha$ (or $\beta$) is large.

## .5.2   MODE RESOLUTION

The precision and recall metrics only take into account the support of original and synthetic data, but not the actual densities. The density and coverage metric do take this into account, but here we show these are not able to capture this well enough to distinguish similar distributions.

In this experiment we look at mode resolution: how well is the metric able to distinguish a single mode from two modes? Let the original distribution be a mixture of two gaussians that are separated by distance $\mu$ and have $\sigma = 1$,

$$X \sim \frac{1}{2}N(-\frac{\mu}{2}, 1) + \frac{1}{2}N(+\frac{\mu}{2}, 1)$$

and let the synthetic data be given by

$$Y \sim N(0, 1 + \mu^2).$$

This situation would arise if a synthetic data generator fails to distinguish the two nodes, and instead tries to capture the two close-by modes of the original distribution using a single mode. We compute metrics for $\mu \in [0, 5]$.

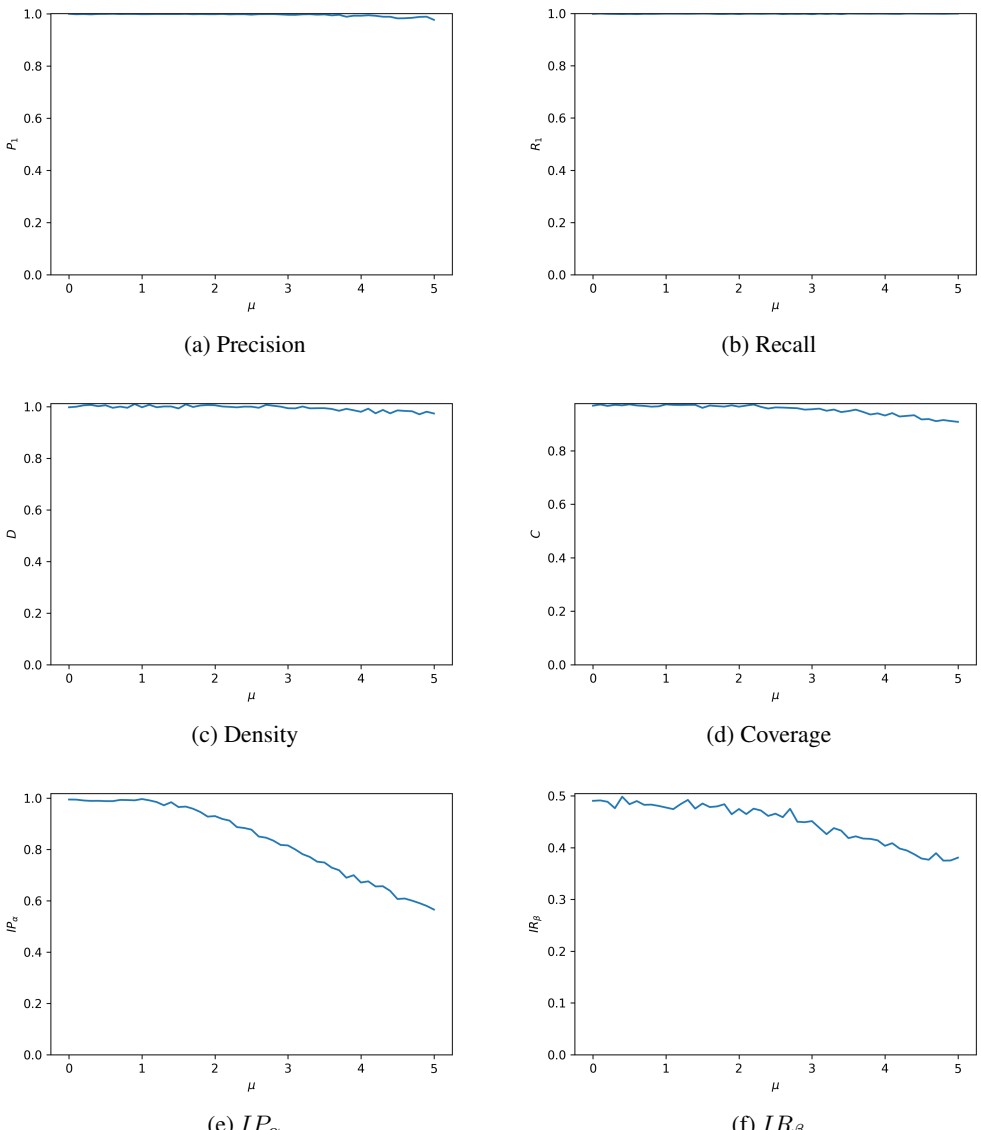

Figure 3: Toy experiment II: mode resolution

As can be seen, neither P&R nor D&C notice that the synthetic data only consists of a single mode, whereas the original data consisted of two. The $\alpha$-precision metric is able to capture this metric: for small $\alpha$ the $\alpha$-support of the original distribution is centred around the two separated, and does not contain the space that separates the modes (i.e. the mode of the synthetic data).

