# OpenReview forum: "How Faithful is your Synthetic Data? Sample-level Metrics for Evaluating and Auditing Generative Models"
_ICLR.cc/2022/Conference — ICLR 2022 Submitted_

### Official Review · Reviewer_qovd · 2021-11-02

**Correctness:** 4
**Technical Novelty And Significance:** 2
**Empirical Novelty And Significance:** 2
**Recommendation:** 8
**Confidence:** 4

**Main Review:**

Strengths:

1. Well written, easy to follow.
2. Timely topic.
3. Discussion of NeurIPS challenge is important.

Weaknesses:

1. I did not learn something substantial from reading the paper.
2. The claims are too strong.

Detailed comments:

1. The generalized definitions of precision and recall are effective markers of understanding synthetic data quality. However, empirically they do not outperform other distribution measuring mechanisms (such as FID, PW, W) and I suspect methods like KL divergence or MMD, or TV distance will also perform comparably.
2. To my understanding, approaches like FID, KL Divergence, MMD, TV distance are also “model agnostic”; I humbly request the authors to rethink the positioning of their work along this particular claim.
3. The appeal in this work lies in the fact that the proposed measures can be calculated on a per-sample basis, unlike prior works.
4. Empirically, the authors consider few issues which may result in poor fidelity/diversity (mode collapse being one of them). If the models did not generalize during training, how do these methods capture them?
5. Could the authors clarify how the proposed method provides interpretability? The authors observe correlations between induced failures and their scores, but they can’t really attribute the score to a particular failure event?
6. The proposed measure for authenticity relies on the ability to detect overfitting. However, recent work by Feldman et al. argues that overfitting (or memorization, in some cases) is essential for generalization. Can the authors comment on the same?


**Summary Of The Paper:**

The authors propose a mechanism to calculate the "goodness" of a generative model on a per-sample level; this method is model agnostic. This can be used to audit the model post-hoc.

**Summary Of The Review:**

Key requirements: I would urge the authors to empirically validate if overfitting has occurred to validate their claims (maybe through MI attacks).

---

> ### Author Response · Authors · 2021-11-17
> **Response to comments by Reviewer qovd**
>
> Thank you for your comments. We provide a point-by-point response below.
>
> 1- **Comparison with distribution-based metrics (KL divergence or MMD, or TV distance).** Please note that the goal of our metrics is not to “outperform” distribution-based metrics, but rather to disentangle the components of distributional discrepancy into a precision component (a count for the fraction of samples that look realistic) and a recall component (a count for fraction of real samples that could be generated by the model). In this sense, our metrics are qualitatively different than distribution-based metrics and can quantify different modes of failure of a generative model for debugging purposes in way that cannot be achieved with distribution-based metrics. This was shown in our MNIST experiment, where we illustrate how the $IR_\beta$ and $IP_\alpha$ were used to diagnose mode collapse in an interpretable fashion that is not possible with the Wasserstein distance and FID distance metric.
>
> 2- **Model agnosticism of other metrics.** The distribution-based metrics (FID, KL Divergence, MMD, TV distance) can be domain-agnostic if Euclidean distances between data points in the ambient feature space describe similarity between samples. In more complex applications where the data lives in a learning manifold, these metrics only make sense if applied on top of pre-trained, application-specific representations. We do not believe we stated that these metrics are not model-agnostic since they can be applied to any model as correctly pointed out by the reviewer. We will highlight this in the final version of the paper.
>
> 3- **What if models did not generalize during training?** If the model is poorly trained or fails to generalize, this will show up in the generated samples being noisy (poor precision) and the coverage of real samples being inadequate (poor recall). In the extreme case when a model does not learn anything from the data, the recall and precision curves will collapse to zero for all values of $\alpha$ and $\beta$.
>
> 4- **How do our metrics provide interpretability?** The proposed metrics provide interpretability in two ways: (I) first the sample level metrics flag certain samples in synthetic data as “low precision” and the recall metric flags real samples that are not covered by the generative model, which can help the modeler understand the failure events of the model at the sample level, and (II) the $\alpha$-precision metric differentiates between synthetic samples that look like typical samples in the true distribution and ones that look like outliers. When the precision is high for low and high values of $\alpha$, then the generative model does a good job capturing the entire real distribution. If the precision value is low for low values of $\alpha$, then the generative model does not properly capture the modes of the real distribution because the modes are concentrated around the $\alpha$ support of the real data, indicating a potential mode collapse. The MNIST experiment is a demonstration for how our metrics can be used to diagnose mode dropping ($IR_\beta$ decreases with dropping probability while $IP_\alpha$ is not affected by this, hence together the metrics give an interpretable way.
>
> Note that our metrics mainly serve as indicators for different modes of failure, but it is impossible to design an evaluation procedure that describes the infinitely many ways a model can be wrong using a finite number of metrics. The key strength of our metrics is that it can help identify much more modes of failure compared to existing metrics.
>
> 6- **Authenticity metric and memorization.** Please note that the work by V. Feldman analyzes discriminative models and not generative models---in predictive models, it might be the case that good generalization performance requires interpolating the input space. In generative models, however, memorization means that the model samples one of the training data points at a much higher frequency than its true density in the real data distribution. It is not known if memorization is necessary for good density estimates in generative modeling. Moreover, our main concern with memorization/overfitting comes from a concern over privacy and not performance. Hence, even if memorization turns out to be necessary for generalization, it would still be useful to spot the memorized samples in the generated data using our metric and exclude them from the synthetic sample to retain privacy of subjects in the real data. This is the **auditing** use case described in Section 2.3. In this case, the memorized sample will be used to train a good generative model, but then these samples will be trimmed off of the model’s output to avoid privacy leakage.

---

> > ### Comment · Reviewer_qovd · 2021-11-18
> > **Thank you for your response!**
> >
> > Thank you, authors, for clarifying my concerns.
> >
> > Please discuss my concerns in the revised version of the paper. The score is updated.

---

### Official Review · Reviewer_AgxQ · 2021-11-03

**Correctness:** 4
**Technical Novelty And Significance:** 2
**Empirical Novelty And Significance:** 2
**Recommendation:** 3
**Confidence:** 5

**Main Review:**

Strengths:
+ Clear writing and demonstrations.
+ Meaningful research topic.
+ Technically reproducible.

Weaknesses:
- The technical novelty is marginally revised from Sajjadi et al. 2018.
- The experiments are incomplete and therefore unconvincing.
  - The testing datasets are a bit toy-like. Like Sajjadi et al. 2018, please also experiment on CelebA [1] or FFHQ [2]. For mode collapse evaluation, please consider using attribute classifiers pre-trained on CelebA, as well as IvOM [3].
  - The testing generative models are out of date. Results on old-fashioned architectures are not conclusive for cutting-edge research works. Imagine our community works on a new GAN paradigm and targets to outperforming StyleGAN3 [1], are we convinced to use the proposed metric which has only been validated on toy data/models? For GAN, please consider validating on StyleGAN3 [4]. For VAE, please consider validating on VQ-VAE-2 [5].

[1] Liu, Ziwei, et al. "Deep learning face attributes in the wild." ICCV 2015.

[2] Karras, Tero, Samuli Laine, and Timo Aila. "A style-based generator architecture for generative adversarial networks." CVPR 2019.

[3] Metz, Luke, et al. "Unrolled generative adversarial networks." ICLR 2017.

[4] Karras, Tero, et al. "Alias-free generative adversarial networks." NeurIPS 2021.

[5] Razavi, Ali, Aaron van den Oord, and Oriol Vinyals. "Generating diverse high-fidelity images with vq-vae-2." NeurIPS 2019.

**Summary Of The Paper:**

This paper targets to a 3-dim metric: alpha-Precision, beta-Recall, and Authenticity, that quantifies the fidelity, diversity, and generalization performance of a generative model. The proposed metric serves as soft-boundary classifiers between real and generated spherical-shaped supports, and improves the robustness against outliers and generation failure cases. This sample-wise metric can audit models by judging individual synthetic samples by their quality.

**Summary Of The Review:**

See the main review.

---

> ### Author Response · Authors · 2021-11-17
> **Response to comments by Reviewer AgxQ**
>
> Thank you for your review. Below we provide a point-by-point response to your comments.
>
> ***Technical novelty***
>
> Please note that while (Sajjadi et al. 2018) proposes precision-recall metrics for evaluating generative models, this does not undermine our contribution since precision-recall analyses is a general framework for model evaluation, and various other previous work belong to the same strand of literature (e.g., Kynkäänniemi et al., 2019 and L Simon et al 2019). Our work is both conceptually and practically different: it is the first to introduce the concept of $\alpha$-volume support for evaluating precision and recall metrics to conduct elaborate comparisons between distributions in an interpretable way, which is not achievable by other metrics. Most notably, the standard precision and recall metrics (i.e., the extreme points on the precision-recall curve by Sajjadi et al. 2018) is only a special case of our precision and recall curves for $\alpha=\beta=1$. In terms of performance, Appendix 5 shows how the original precision and recall metrics are not robust to outliers and don’t capture mode resolution, hence our metrics are a clear improvement here. This is also shown in the MNIST mode dropping experiment, in which $IR_\beta$ is more sensitive to mode collapse than the original recall.
>
> ***Further experimental results***
>
> Thank you for these suggestions.
>
> Including further experiments on CIFAR-10 and CelebA with larger generative models such as StyleGAN and DDPM is going to enrich our paper. We are going to add such experiment to the supplementary material. Please note that the performance of our metric (and all other metrics) is oblivious to the complexity/sophistication of the generative model, in the same way the accuracy of an AUC-ROC score is independent of the complexity of a classification model---these metrics only operate on samples. The experiments also dealt with high-dimensional data: the time series data in (Sec. 5.3) involves thousands of features per patient over time, and for MNIST, we scaled up images and used an InceptionV3 network embedding (2048 dim); the same embedding that would be used for higher quality images.
>
> We strongly believe that our evaluation is very comprehensive—unlike previous works, it involves different data types (tabular, images, time-series) and demonstrates all key advantages and use cases of our metric. Since a key premise of our paper is domain-agnosticism, we chose to diversify data types in our experiments rather than restricting to image synthesis.
>
> Based on your suggestions, we have obtained new results on the CIFAR-10 data set. In these new experiments, we compared SOTA generative models (DDPM and StyleGAN) models using their open-source implementations in [R1] and [R2]. For both models we created 10,000 CIFAR-10 images using the included pre-trained models. Subsequently, both synthetic datasets were evaluated in comparison to the original CIFAR10 test set. For the one-class representations, we first embedded all images using a pretrained InceptionV3 model. The preliminary results for the comparison between the two models using the $IP_\alpha$ and $IR_\beta$ metrics is summarized below:
>
> Model-----|          $IP_\alpha$        |          $IR_\beta$          |
> __________________________________________________
>
> StyleGAN    |            0.858              |            0.663           |
>
> DDPM.        |            0.921              |            0.501           |
>
> While the FID metric ranks StyelGAN higher than DDPM, we found that DDPM captures the distribution of the data better in the region where the supports of the two models overlap, as evident by the superior IPα of DDPM. Our analysis paints a more complex picture for the comparison between both models: StyleGAN scores higher in the recall metric, indicating a better capturing of the diversity of images.
>
> In the final version of the Appendix, we will add this experiment along with the detailed $P_\alpha$ and $R_\beta$ curves analyzing the ways in which DDPM and StyleGAN perform well and the different ways in which they fail. We will also add the other experiments requested in the Appendix.
>
> **References**
>
> *[R1] Song, Yang, and Stefano Ermon. "Generative Modeling by Estimating Gradients of the Data Distribution." Proceedings of the 33rd Annual Conference on Neural Information Processing Systems. 2019.*
>
> *[R2] Karras, Tero, Samuli Laine, and Timo Aila. "A style-based generator architecture for generative adversarial networks." Proceedings of the IEEE/CVF Conference on Computer Vision and Pattern Recognition. 2019.*

---

> > ### Comment · Reviewer_AgxQ · 2021-11-30
> > **Thank you for your response!**
> >
> > Thanks for the authors’ careful response! After reading the response and the other reviews, I will have to **keep my “reject” score**. Since this score differs from the others, I am trying to articulate my opinions in a clear logic.
> >
> > In fact, none of the critical concerns in my initial review is addressed properly, which leads to the same conclusion of **Reviewer caMn: “I am not fully convinced this is a practically useful paper”**. In particular, my major concerns stem from the fact that **the testing datasets are toy-like, and the testing models are out of date.**
> > - The authors agree on the necessity to test on more practical datasets and more recent generative models that “Including further experiments on CelebA with larger generative models such as StyleGAN is going to enrich our paper.” However, too many promises like “We are going to add such experiment to the supplementary material” or “In the final version of the Appendix, we will add this experiment along with…” confirm this submission is half-baked. The next iteration with a resubmission is highly encouraged.
> > - The authors’ claim of “we chose to diversify data types in our experiments rather than restricting to image synthesis” does not touch the point. Diversification is good, but it cannot be a compromise of not testing on the recent state-of-the-art of each data domain. In the image domain, it should be StyleGAN2 or StyleGAN3 on a regular resolution (at least 128x128). WGAN-GP on MNIST or CIFAR-10 with a resolution smaller than 32x32 is far behind the current standard. Proposing a new metric is a serious mission. Once it is accepted, it should be expected to impact the current benchmarking protocol. Yet this work cannot reach this standard because it is never validated on the current benchmarks.
> > - The authors’ claim of “the performance of our metric (and all other metrics) is oblivious to the complexity/sophistication of the generative model” is not well-supported. If it was valid and if the authors stood on the state-of-the-art perspective, the proposed metric should have been naturally tested on the SOTA (which are the most recent sophistications), not just on toy-like settings. It is fine to start with toy-like settings and to motivate this line of research, but they cannot be the sharp end of a paper like the current version.
> > - “Based on your suggestions, we have obtained new results on the CIFAR-10 data set.” Unfortunately this is not my suggestion. CIFAR-10 is at 32x32 resolution which is not practically useful. My suggestions are summarized again as below which were not addressed by the authors' response:
> > 1. Experiment on CelebA with at least 128x128 resolution, or even FFHQ with its full 1024x1024 resolution.
> > 2. For GAN, experiment on StyleGAN2 or StyleGAN3. For VAE, experiment on VQ-VAE-2.
> > 3. To quantitatively evaluate mode collapse, using attribute classifiers pre-trained on CelebA as well as IvOM would make the authors’ claim of “We strongly believe that our evaluation is very comprehensive” more convincing.

---

### Official Review · Reviewer_caMn · 2021-11-03

**Correctness:** 3
**Technical Novelty And Significance:** 3
**Empirical Novelty And Significance:** 3
**Recommendation:** 6
**Confidence:** 3

**Main Review:**

The paper contributes useful thought and discussion for how best to argue the holistic performance of a generative model, contrasting with FID and other domain-specific approaches.  There are novel perspectives in this paper that are worth sharing with the community, particularly those contributing to discussion around generalization vs. memorization (copying) where it is shown that prior metric ignoring this aspect otherwise indicate performant models (c.f. time-series NeurIPS challenge data).  The paper could be improved by drawing deeper relationship with prior thought in this area e.g. Adlam et al. (2019), Meehan et al. (2020) who are cited in Sec 3.2.2 but not discused.  Indeed the brief literature review is relegated to the paper Appendix and was not helpful in arguing for the novel contribution of the paper in context of such work.  The paper also lacks any reflection or conclusion on the limitations of the proposed metrics.  It was unclear to me how the proposed authenticity classifier would work for multi-modal distribution given the assumptions of noise within a hypersphere.  It did not seem like it would be practical for realistic, complex data problems.  A use case on high-dimensional data e.g. images beyond a toy MNIST example would have better demonstrated the practical utility of this work for data where the need of such performance metrics is clearer.

**Summary Of The Paper:**

The paper presents a methodology assessing the performance of a generative model in a domain agnostic fashion.  A 3-dimensional metric space is proposed: fidelity (output quality), diversity (coverage of expected variability of output) and generalization (to what degree model avoids memorizing training data i.e. is truly generative) is proposed.  Particularly the latter element is novel and the manner in which it is may be evaluated by developing an `authenticity’ measure for the task. Three illustrative use cases in image (mnist) and medical patient data (COVID-19, `Hide and Seek’ seq2seq data) domains are provided.

**Summary Of The Review:**

I am on the borderline tending toward accepting the paper given the interesting discussion around authenticity and memorization, but I am not fully convinced this is a practically useful paper.

---

> ### Author Response · Authors · 2021-11-17
> **Response to comments by Reviewer caMn**
>
> Thank you for your review.
>
> **Regarding the connection with the work by Adlam et al. (2019), Meehan et al. (2020).** We referred to these works in the context of explaining how measuring generalization in generative models. The work by (Adlam et al. (2019)) is limited to Wasserstein GAN models, and does not propose a metric for evaluating overfitting but rather uses discriminators unseen by the generator to measure generalization. The work in (Meehan et al. (2020)) is relevant to our work as the authors also develop a hypothesis test for data copying. However, this work tests the "global" hypothesis that a distribution $P$ is copied from $Q$ and does not apply a test on the sample-level to derive a performance metric. We will elaborate on the connections between our metrics and these works in the revised version of the paper.
>
> We believe that the novelty of our contribution was demonstrated clearly through the elaborate illustration of how our metrics differ from the standard precision-recall metrics in Figure 3, and how the proposed metrics can offer more use cases and provide more detailed diagnostics of generative models in Section 5. In the context of previous work on memorization/over-fitting/data-copying, our authenticity metric is the first optimal sample-level hypothesis test for data-copying (according to the Neyman–Pearson lemma).
>
> The authenticity classifier relies on a non-parametric distance-based measure to construct the test statistic, hence it does not rely in any way on the distribution being unimodal or multimodal. The hyper-spheric construction is only needed to evaluate the $\alpha$- and $\beta$-supports for the precision and recall metrics.
>
> Finally, please note that the experimental evaluations involved very high dimensional data sets such as the time-series NeurIPS challenge which comprises data points with thousands of dimensions (time steps and feature dimensions per time step).
>
> Thanks again for your review and we are looking forward to your feedback during the discussion period.

---

### Official Review · Reviewer_hx4h · 2021-11-10

**Correctness:** 4
**Technical Novelty And Significance:** 3
**Empirical Novelty And Significance:** 2
**Recommendation:** 6
**Confidence:** 4

**Main Review:**

Strengths: The paper addresses an important topic that is time critical and relevant to the generative modeling community and the broader sphere of AI. It is well written and motivated and a pleasure to read overall. The authors clearly point out the issues that are lacking with existing metrics and motivate the need to define the new versions of precision / recall metrics. Moreover, the metrics enable a new use case of model auditing which can be very relevant is safety critical applications.

Weaknesses:
1. The paper promises a lot in terms of the defined metrics but the evaluations are bit underwhelming. The experiment with the predictive model depends on the downstream classifier as well - the performance of the generative model is tied to the performance of the classifier - hence the hyper-parameter settings and training settings for the classifier should be explained clearly.
2. Another aspect that is missing from the evaluation is the application of model auditing on real life critical data. The idea of model auditing becomes useful mainly in cases where we need to filter at the sample level. When a generative model is trained on medical data (imaging) how do the metrics help in model auditing?
3. An interesting question to probe is how to compare different generative models that have comparative expertise? For instance, one model could be an expert in color and another could be in shape and so on - can we use the defined metrics to identify such behaviors?
4. Please provide error bars on the predictive modeling experiments - as in, account for the randomness in the generative model training process.

**Summary Of The Paper:**

This work addressed an interesting problem and a very relevant one - how to audit generative models? The most popular metric to evaluate generative models is FID but it is extremely opaque and population based. This paper proposes new fidelity metrics that overcome both these challenges. While the definition of these metrics are well grounded, their measurement process itself - which is done via trained NN classifiers - is a bit unconvincing. The authors demonstrate that the proposed metric can recover the real ranking of generative models on a synthetic data generation task and can discover issues like mode collapse.

**Summary Of The Review:**

Overall, the paper does a good job in addressing a relevant problem in a well motivated manner. Even though there are some open concerns regarding the evaluation, the fact that the current approach is one of the first to facilitate model auditing for generative models makes this a good contribution. Authors, please address my concerns in the weaknesses listed above.

---

> ### Author Response · Authors · 2021-11-17
> **Response to comments by Reviewer hx4h**
>
>
> Thank you for your excellent review. Below is a point-by-point response to your comments.
>
> 1. **Experiments** We believe that the experiments covered many applications, data types and use cases within the limited space we have in the submission, and more experiments were provided in the Appendix. It is true that the  predictive modeling experiment depends on the classifier, but this will be the case in a real-world predictive modeling use case. To minimize the impact of this dependence on our comparison of metrics, we selected a simple logistic regression model that has no hyper-parameters. The hyper-parameters of the generative models are irrelevant since our goal is to compare metrics of performance applied to these models, and not to establish the superiority of any model. Hence, we believe that the predictive modeling experiment is very representative of what a real-world use case would like and is accurate in its evaluation of the various performance metrics.
>
> 2. **Model auditing on real life critical data** We agree with you that model auditing is useful mainly in cases where we need to filter critical data (such as medical data) at the sample level. However, we do not believe that this is missing from the evaluation. In fact, this is exactly what Figures 4-(b) and 4-(c) demonstrate. As you rightly mentioned, because our metrics are defined on the sample level, we can discard unauthentic or imprecise samples. In the experiments in Figures 4-(b) and 4-(c) , we show how post-hoc model auditing can be applied to COVID-19 patient data that includes sensitive information to improve the quality of synthetic data generated by an already-trained model. In these Figures, we show that model auditing does not only lead to nearly optimal precision and authenticity for the curated data (Figure 4(c)), but also improves the AUC-ROC of the predictive model fitted to audited data (from 0.76 to 0.78 for the audited ADS-GAN synthetic data, p < 0.005), since auditing eliminates noisy data points that would otherwise undermine generalization performance.
>
> 3. **Generative models with comparative expertise** Thank you for bringing up this point. This is indeed a very interesting use case for our metrics. Because they are defined on the sample level, we can identify the type of samples that the model can precisely generate, and the type of samples that the model is missing from the real data using the sample-level binary indicators of the precision and recall metrics. Once we identify which samples each generative models get right and which ones are missed, we can analyze these samples to see the common features in them. This is possible because our metrics are both sample-level and model-agnostic, so they can compare any pair of generative models with respect to the individual data points they generate.
>
> 4. **Error bars** Thank you for this suggestion. We have calculated p-values instead and all results in Figure 4 are statistically significant. We will convert the p-values to confidence intervals and add error bars in the final paper.

---

### Decision · Program_Chairs · 2022-01-20

**Decision:**

Reject

**Comment:**

The authors propose a new set of metrics for evaluation of generative models based on the well-established precision-recall framework, and an additional dimension quantifying the degree of memorization. The authors evaluated the proposed approach in several settings and compared it to a subset of the classic evaluation measures in this space. The reviewers agreed that this is an important and challenging problem relevant to the generative modeling community at large. The paper is well-written and the proposed method and motivation are clearly explained.

The initial reviews were borderline, and after the discussion phase we have 2 borderline accepts, one strong accept, and one strong reject. After reading the manuscript, the rebuttal, and the discussion, I feel that the work should not be accepted on the grounds of insufficient empirical validation. Establishing a new evaluation metric is a very challenging task -- one needs to demonstrate the pitfalls of existing metrics, as well as how the new metric is capturing the missing dimensions in a thorough empirical validation. While the former was somewhat shown in this work (and in many other works), the latter was not fully demonstrated. The primary reason is the use of a non-standard benchmark to evaluate the utility of the proposed metrics. I agree that covering a broader set of tasks and models makes sense in general, but it shouldn’t be done at the cost of existing, well-understood benchmarks. I expected to see a thorough comparison with [1], one of the most practical metrics used today which can be easily extended to all settings considered in this work (notwithstanding the drawbacks outlined in [2]). What are the additional insights? What is [1] failing to capture in practical instances? Does the rank correlation change with respect to modern models across classic datasets (beyond MNIST and CIFAR10)? This would remove confounding variables and significantly strengthen the paper.

My final assessment is that this work is borderline, but below the acceptance bar for ICLR. I strongly suggest the authors to showcase the additional improvements over methods such as [1] in practical and well-understood settings commonly used to benchmark generative models (e.g. on images). The experiments suggested by the reviewers are a step in the right direction, but not sufficient.

[1] Improved Precision and Recall Metric for Assessing Generative Models. Tuomas Kynkäänniemi, Tero Karras, Samuli Laine, Jaakko Lehtinen, Timo Aila. NeurIPS ’19

[2] Evaluating generative models using divergence frontiers.
Josip Djolonga, Mario Lučić, Marco Cuturi, Olivier Frederic Bachem, Olivier Bousquet, Sylvain Gelly. AISTATS ‘20